# Helios as a Potential Biomarker in Systemic Lupus Erythematosus and New Therapies Based on Immunosuppressive Cells

**DOI:** 10.3390/ijms25010452

**Published:** 2023-12-29

**Authors:** Andrés París-Muñoz, Odelaisy León-Triana, Antonio Pérez-Martínez, Domingo F. Barber

**Affiliations:** 1Department of Immunology and Oncology and NanoBiomedicine Initiative, Centro Nacional de Biotecnología (CNB-CSIC), 28049 Madrid, Spain; aparis@ext.cnio.es; 2Translational Research in Pediatric Oncology, Hematopoietic Transplantation and Cell Therapy, IdiPAZ, Hospital Universitario La Paz, 28049 Madrid, Spain; odelaisy.leon@idipaz.es (O.L.-T.); aperezmartinez@salud.madrid.org (A.P.-M.); 3IdiPAZ-CNIO Pediatric Onco-Hematology Clinical Research Unit, Spanish National Cancer Research Centre (CNIO), 28049 Madrid, Spain

**Keywords:** Helios, biomarkers, systemic lupus erythematosus, autoimmunity, tolerogenic DC, nanoparticle-mediated magnetic targeting

## Abstract

The Helios protein (encoded by the *IKZF2* gene) is a member of the Ikaros transcription family and it has recently been proposed as a promising biomarker for systemic lupus erythematosus (SLE) disease progression in both mouse models and patients. Helios is beginning to be studied extensively for its influence on the T regulatory (Treg) compartment, both CD4^+^ Tregs and KIR^+^/Ly49^+^ CD8^+^ Tregs, with alterations to the number and function of these cells correlated to the autoimmune phenomenon. This review analyzes the most recent research on Helios expression in relation to the main immune cell populations and its role in SLE immune homeostasis, specifically focusing on the interaction between T cells and tolerogenic dendritic cells (tolDCs). This information could be potentially useful in the design of new therapies, with a particular focus on transfer therapies using immunosuppressive cells. Finally, we will discuss the possibility of using nanotechnology for magnetic targeting to overcome some of the obstacles related to these therapeutic approaches.

## 1. General Considerations on the Immune System

The immune system of complex organisms like mammals is a coordinated network of innate and adaptive cells, molecules, tissues, and organs that has evolved to fight against the foreign pathogens causing infectious disease. In parallel, the immune system also has the means to protect host self-antigens, part of a dynamic homeostatic equilibrium aimed at preventing any potential side effects due to an overzealous immunogenic response.

Unlike innate cells, one of the essential features of adaptive cells is their ability to recognize a wide range of different antigens (endogenous and exogenous). Adaptive immunity is driven by cellular (helper or CD4^+^ T cells, and cytotoxic or CD8^+^ T cells) and humoral immunity (via B cells). From a functional and perhaps simplified perspective, each immune response (innate and adaptive) can be classified as immunogenic (e.g., the immune response directed against pathogenic microorganisms) or tolerogenic (for example, the immunosuppression of autoreactive processes). Overall, the immune system aims to establish a dynamic and continuously regulated balance between these two opposing forces: reactivity to foreign molecules and self-antigen tolerance [1,2]. Interestingly, the same cell type may be involved in both activities, depending on factors like the molecular environment, their activation, and the patterns of gene expression [3]. Most of the known innate and adaptive immune cell populations are presented in Figure 1, showing their immunogenic and tolerogenic counterparts. Finding elements common to both sides of the balance, and that influence the general status of the immune system, will be interesting to better understand pathologies with a predominantly immune component, such as cancer and autoimmunity. In this review, we will discuss the possibility of considering the Helios transcription factor (encoded by the *IKZF2* gene) as a potential biomarker and therapeutic target in systemic lupus erythematosus (SLE), a relatively common autoimmune disease.

### 1.1. The Adaptive Immune Equilibrium

In addition to T cells, dendritic cells (DCs) play a crucial role in lymphoid organs (thymus, lymph nodes, and spleen), orchestrating the adaptive immune system. In this sense, here we will review the role and identity of the different tolerogenic and immunogenic immune subpopulations, focusing on T cell and DC repertoires.

#### 1.1.1. Conventional CD4^+^ and CD8^+^ T Cells

After their generation in bone marrow from hematopoietic progenitors, in both humans and mice, immature T cells move to the thymus to complete their development. Once there, and during their maturation, thymocytes reorganize their TCR by genetic recombination to generate a specific repertoire of immune cells with distinct affinities for the MHC molecules associated to their peptides. Subsequently, after positive and negative selection, the remaining fraction of TCRαβ^+^CD4^+^CD8^+^ cells silence one of their co-receptors, CD8 or CD4 depending on the interaction with antigen presenting cells (APCs), through their MHCI or MHCII molecules, respectively [4,5]. Thus, naive TCRαβ^+^CD4^+^CD8^−^ (restricted MHCII cells) and TCRαβ^+^CD4^−^CD8^+^ (restricted MHCI cells) T lymphocytes arise in the thymus, exhibiting tolerance for self-antigens towards peripheral lymphoid tissues to be activated by APCs (mainly, DCs), and they then trigger the cellular adaptive response. During this process, T cells undergo some notorious changes in the expression of several well-known immune markers, from naïve to effector, and to central or effector memory, which can easily be analyzed by flow cytometry (Table 1) [6,7].

Globally, these mature T lymphocytes are defined as conventional T cells (Tconv), and they are responsible for the positive and immunogenic responses to foreign antigens. Yet what are the homeostatic mechanisms that avoid the potentially autoreactive reactions of Tconv cells that escape the process of negative selection in the thymus? Other subpopulations of immune cells play a pivotal role in readdressing the immune equilibrium through immunosuppression, cells known as T regulatory cells (Tregs). Among these cells, most research has focused on CD4^+^ Tregs, and in particular, on a CD4^+^ Treg population derived from the thymus (CD4^+^ tTreg). These cells survive the process of negative selection and they exhibit high affinity for MHCII-self-peptide complexes, expressing higher levels of CD25 and FoxP3 [8]. Nevertheless, incipient CD8^+^ Tregs that express the receptors of natural killer (NK) cells are currently generating much interest [9].

#### 1.1.2. Regulatory CD4^+^ T Cells

The discovery of the FoxP3 transcription factor as a marker and an essential regulator that maintains CD4^+^ tTreg’s immunosuppressive activity implied a paradigm shift in the field of immunology [10,11]. After years of intense research in this topic, a CD4^+^ tTreg population (CD25^+^ FoxP3^+^) has been clearly shown to maintain the peripheral immune equilibrium by controlling aberrant and exacerbated autoreactive responses against self-antigens, microorganisms, and environmental antigens [12,13]. In this regard, several allergies and autoimmune pathologies are associated with certain alterations to CD4^+^ tTregs, revealing the true importance of their correct function [14]. In addition, some roles of CD4^+^ tTregs in other body systems beyond the immune system have also been reported. For example, the close links between these lymphocytes and tissue regeneration [15], or other metabolic diseases like obesity [16], have been studied.

Although CD4^+^ tTreg lymphocytes represent the most abundant fraction of Tregs in the periphery, there are also other regulatory CD4^+^ cell types of different origins that reliably express FoxP3 and that participate in establishing tolerance. For example, peripheral CD4^+^ Tregs (CD4^+^ pTregs) are generated from CD4^+^ Tconvs outside the thymus in tolerogenic contexts, such as in the intestinal mucosa. Moreover, CD4^+^ iTregs can be induced in vitro from CD4^+^ Tconvs under specific culture conditions and these cells are currently being tested in clinical trials as potential autologous cell therapy for patients affected by several autoimmune diseases [17]. Despite their differences, these CD4^+^ tTregs, pTregs, and iTregs employ some common immunosuppressive mechanisms.

Given the intrinsic molecular and cellular complexity of the immune system, CD4^+^ Treg lymphocytes need to act on several elements and processes, of both the innate and adaptive responses, in order to ensure effective peripheral tolerance in a coordinated manner [18,19,20] (see Figure 2 for a summary of the main immunosuppressive strategies identified in CD4^+^ Tregs). Among these events, the overexpression of CD25 (the α subunit of the interleukin 2 receptor or IL-2R) facilitates the capture and sequestering of the available IL-2. The reduced availability of IL-2 specifically dampens the activation and proliferation of CD8^+^ Tconv cells that depends on this cytokine [21]. From a metabolic perspective, the surface ectoenzymes CD39 and CD73 transform extracellular ATP into adenosine molecules, the latter potentially inhibiting effector T and myeloid cells (DCs and macrophages) [22,23]. Furthermore, these Tregs produce tolerogenic cytokines like IL-10 and TGF-β that exert multiple effects on immune cells. By contrast, IL-10 and TGF-β inhibit T and B lymphocyte proliferation and stimulation [24]. Moreover, these cytokines would polarize DCs towards a more tolerogenic phenotype, which promotes the induction of pTregs [25]. Finally, Tregs and DCs interact directly through two different signaling systems that could induce a tolerogenic state in DCs: PD-1 (Treg)—PDL-1 (DCs) [26], and CTLA-4 (Treg)—CD80/86 (DCs) [27].

#### 1.1.3. Regulatory CD8^+^ Tregs

Globally, although most biomedical research into Treg cells has focused on the CD4^+^ Treg compartment, a recently described subset of CD8^+^ Treg lymphocytes that expresses inhibitory markers from NK cells is being investigated intensively in mice and humans. Previously, a wide range of different CD8^+^ T cells exhibiting immunomodulatory properties were classified as CD8^+^ Tregs, yet with heterogeneous phenotypes: CD8^+^ FoxP3^+^ [30], CD8^+^ CD28^−^ [31], CD8^+^ CD103^+^ [32], and CD8^+^ CD122^+^ CD49d^low^ [33]. However, unlike CD4^+^ tTreg cells, these populations were found peripherally in very small proportions or in experimental contexts of antigenic exposure in mice. In other words, these subsets were never found naturally in significant numbers in healthy and young mice without prior manipulation [34,35].

This situation changed with the discovery of a new regulatory subpopulation of CD8^+^ T cells in naïve mice without any immune disruption, reliably defined by the expression of three markers: CD44^+^, CD122^+^, and Ly49^+^ (Ly49^+^ CD8^+^ Tregs from here on). This population has the ability to modulate the adaptive immune response by eliminating autoreactive CD4^+^ Tconv and follicular T lymphocytes [36,37,38]. Since then, other properties of this population have been clarified, such as (1) the absence of intranuclear FoxP3 expression; (2) high levels of surface CD127 (in contrast to CD4^+^ Treg cells) [38]; (3) homeostatic regulation, mainly through IL-15 and TGF-β [39,40,41]; (4) phenotypic acquisition induced peripherally after thymic maturation [42]; and (5) finally, the fact that these cells have a non-redundant role in controlling autoreactive antibody titers by acting in germinal centers from lymphoid organs [37,41], thereby regulating autoimmune phenomena [40,41,43]. In terms of their functionality, the best-defined effector mechanism used by CD8^+^ Ly49^+^ Tregs is the cytotoxic killing of autoreactive CD4^+^ T cells, regardless of IL-10 production [36] (Figure 2).

#### 1.1.4. Tolerogenic Dendritic Cells (tolDCs)

As previously stated, DCs are the primary APCs that initiate the adaptive immune responses of T lymphocytes in response to foreign antigens in the periphery. However, in addition to this crucial role, they also contribute to immune tolerance in the organism in two other ways. Firstly, DCs mediate the negative selection and differentiation of tTregs in the thymus (central tolerance), while they also maintain peripheral tolerance by elimination, anergic induction, or conversion of autoreactive T cells to pTregs [44]. While immature DCs (imDCs) were simply believed to perform a tolerogenic function and mature DCs (mDCs) an immunogenic function, increasing experimental evidence suggests that this functional divide extends beyond the maturation state of DCs. In fact, maturation is necessary for their optimal tolerogenic function, even under basal conditions with no exogenous stimulation. Thus, we will refer to them functionally as imDCs, mDCs, and tolerogenic mature DCs (tolDCs) [45,46].

Apart from these states of maturation, distinct DC subpopulations fulfill specific roles in the immune response. These subpopulations are well-defined in both mouse models and humans, and include conventional (cDCs) and plasmacytoid dendritic cells (pDCs) [47] (Table 2). Type 1 (cDC1s) and type 2 (cDC2s) DCs are defined by their developmental transcriptional program, presenting MHCI antigens for CD8^+^ T cells or MHCII for CD4^+^ T cells, respectively. Furthermore, pDCs specialize in the production of type I interferons upon exposure to viral antigens [48], as well as helping to establish immune tolerance [49]. In line with this, it seems that cDC1 cells, with their inherent capacity to present antigens via MHCI molecules, also play a crucial role in ensuring peripheral tolerance to self-antigens [44,46,50,51,52,53].

Focusing on their tolerogenic functions, tolDCs capture and process self-antigens, whether soluble or derived from apoptotic bodies, for their presentation to peripheral T cells [28]. Among the numerous mechanisms described, some are worthy of particular attention (Figure 2): (1) tolDCs promote the generation of CD4^+^ pTregs from CD4^+^ Tconvs in two ways, by secreting immunosuppressive cytokines (IL-10 and TGF-β) or by signaling through PDL-1 molecules [54,55]; (2) tolDCs eliminate autoreactive CD8^+^ T cells via the Fas/FasL apoptotic system [56]; and (3) tolDCs favor an anergic state in T cells due to the weak expression of MHCII and CD80/CD86 [57].

Also, as for iTreg lymphocytes, it is also possible to induce an immunosuppressive state in mouse and human DCs in vitro for their use in cell-based immunotherapies. This polarization towards tolerance is achieved using different immunomodulators, including glucocorticoids (e.g., dexamethasone), cytokines (e.g., IL-10 and TGF-β), rapamycin, or vitamin D3 [28].

### 1.2. Autoimmune Alterations in the Adaptive Immune Balance

In the case of cancer, the immune balance is inclined in favor of immunosuppression as a strategy to help the tumor evade active immune responses. By contrast, in the case of autoimmunity, the immune equilibrium is oriented towards immunogenicity against self-antigens. In this respect, there is considerable evidence of defects and alterations in number, frequency, and immunosuppressive function of CD4^+^ Tregs from patients with allergies, and other inflammatory or autoimmune diseases [13,14,58]. Although some inconsistencies have been noted depending on the patient cohort, the pathological state, and the different flow cytometry panels employed to define and phenotype Treg subsets, some common defects have been reported globally in CD4^+^ T lymphocytes in autoimmune contexts: (1) the conversion of CD4^+^ Tregs into effector CD4^+^ Tconvs; (2) the reduced ability of CD4^+^ Tregs to promote tolerogenic effects; and (3) the resistance to immunosuppression observed in some activated CD4^+^ Tconvs. By contrast, in type I diabetes, autoreactive CD8^+^ Tconv cells have been extensively described as an important inducer of cytotoxicity of pancreatic β cell death [59], and implicated in other autoimmune pathologies [60]. There is an increase in the number of CD8^+^ KIR^+^ Tregs in both blood and at inflammatory sites in autoimmune patients, and it was proposed that these lymphocytes eliminate pathogenic CD4^+^ T cells to avoid their exacerbated response [9].

Finally, the role of DCs in autoimmune progression has also been analyzed in depth [61,62], establishing that they promote an immunogenic environment due to inflammatory cytokine production. For example, IL-6, IL-12, and IL-23 secretion by immunogenic mDCs facilitates the polarization of self-reactive T cells. In the case of SLE, type-I interferons (IFN-I) mainly produced by pDCs deserve a special mention because they represent one of the most important molecular markers for this disease (see Section 4.4.2).

In summary, the same adaptive cell type can present either tolerogenic or immunogenic properties depending on their phenotype, the stimuli to which they are exposed, and their molecular context. The final cell outcome will depend on different signaling pathways that culminate in gene expression patterns maintained by specific transcription factors. In this regard, it is important to determine if there are any T linage-related transcription factors that contribute to the immunogenic or tolerogenic activity of CD8^+^ and CD4^+^ T cells.

## 2. Helios in the Immune System

One element in this delicate immune balance is the transcription factor Helios (*Ikzf2* in mice and *IKZF2* in humans), a member of the Ikaros family of transcription factors that appears to fulfill a critical role in maintaining a regulatory profile in the T-cell lineage. Helios deficiency, either in mice lacking Helios in all cells [38] or specifically in FoxP3^+^ cells [63] (mostly CD4^+^ Tregs), leads to an autoimmune phenotype at 5–6 months of age that is associated with distinct symptoms, reproducing some canonical immunological manifestations of lupus: hypergammaglobulinemia against nuclear antigens, splenomegaly, glomerulonephritis, an enhanced presence of activated Tconvs, and an expansion of follicular T lymphocytes (see below). Indeed, a germline mutation in *IKZF2* was recently described in an SLE patient that was responsible for severe immune dysregulation [64]. This raises the question as to what the role of Helios is in each immune cell population.

### 2.1. Helios in CD4^+^ T Cells

Initially, Helios was identified in mouse models as a specific marker of CD4^+^ tTregs through a string of discoveries. (1) Until the first week of life, nearly all Tregs (CD4+ FoxP3+ T lymphocytes) that migrate from the thymus express Helios. (2) FoxP3^+^ Helios^−^ CD4^+^ T cells only appear from the second week after weaning. (3) In adult animals, around 70% of the CD4^+^ Treg compartment expresses Helios. (4) Helios is not expressed in either iTregs or pTreg lymphocytes in an experimental tolerance model [65,66]. However, the validity of this latter assumption has become increasingly controversial due to the more recent detection of Helios in CD4^+^ pTregs [67], in activated and exhausted CD4^+^ Tconvs [68,69,70], and in iTregs [71]. Nonetheless, few studies have assessed the relative expression of Helios in specific subpopulations of immune cells beyond simply distinguishing between Helios^+^ and Helios^−^ cells. In this sense, the merits of distinguishing up to three biologically relevant levels of Helios expression is currently being explored (Helios^high^, Helios^mid^, and Helios^low^) [72,73].

Beyond its proposed function as a unique molecular marker for tTregs, it is apparent that Helios^+^ and Helios^−^ CD4^+^ Treg lymphocytes have notable phenotypic, genetic, and functional differences [74,75]. CD4^+^ Tregs expressing Helios exhibit a more activated phenotype, with a higher percentage of effector (CD44^+^ CD62L^−^) cells. Moreover, in vitro studies indicate that these Tregs have a stronger immunosuppressive capacity relative to CD4^+^ Tregs lacking Helios [76]. Furthermore, experiments on lymphopenic animals have shown that CD4^+^ Tregs expressing Helios in vivo express FoxP3 more stably. Indeed, the combined use of FoxP3 and Helios has been proposed to identify bona fide CD4^+^ Tregs in humans [77].

Interestingly, the deletion of Helios in mice, both in all cells [38] and specifically in FoxP3^+^ cells [63], provokes an autoimmune phenotype at 6 months of age, demonstrating the role of this transcription factor in stabilizing regulatory activity. However, the absence of Helios in the entire CD4^+^ T cell repertoire, including both Tconvs and Tregs, does not apparently evoke autoreactivity [38,65]. This curiosity may suggest that Helios is involved in both sides of the immune balance and it leads us to draw two hypotheses. (1) Helios expression in CD4^+^ Tconvs plays a significant role in the autoimmune reaction; and (2) other cell types with Helios-dependent regulatory activity impede autoreactive immune responses.

### 2.2. Helios in KIR^+^/Ly49^+^ CD8^+^

If there is one thing that CD4^+^ Tregs and CD8^+^ Ly49^+^ Tregs have in common, it is the expression of Helios as a maker of their regulatory identity. Nevertheless, research on CD8^+^ Ly49^+^ Tregs is still in its infancy compared to that on CD4^+^ Tregs. Evidence for the physiological relevance of Helios in CD8^+^ Ly49^+^ Tregs and their immunosuppressive function comes from the inability of CD8^+^ Ly49^+^ Helios^−^ T lymphocytes to inhibit the follicular T cell response relative to CD8^+^ Ly49^+^ Helios^+^ cells. Moreover, unlike their Helios^+^ counterparts, CD8^+^ Ly49^+^ Helios^−^ Treg lymphocytes exert a stronger effector phenotype (CD127^low^) and worse survival under inflammatory conditions [38]. Independently, weaker Helios expression has also been found in the CD8^+^ Ly49^+^ Tregs, correlating with autoimmune progression in a murine model that replicates the typical SLE symptomatology from the age of 5 months (including splenomegaly and germinal center reactivity). From this model, with deficient TGF-β signaling, expression of this cytokine may influence the Helios expressed by CD8^+^ T cells [41]. In addition, we found that both the reduction in the proportion of Ly49^+^ CD8^+^ T cells and the weaker Helios expression by these cells was correlated to disease progression in two mouse models of lupus (MRL/MPJ and MRL/LPR) [73].

A recent article in Science characterized the function and phenotype of the human equivalent of Ly49^+^ CD8^+^ Tregs in infectious and autoimmune contexts [9]. Although the murine family of Ly49 receptors lacks a genetic human homologue in terms of sequence, the members of the human Killer cell Immunoglobulin-like Receptor (KIR) family are functionally equivalent to these. Both populations, KIR^+^ CD8^+^ and their murine Ly49^+^ CD8^+^ T cell counterparts, can eliminate autoreactive CD4^+^ T cells in autoimmune and viral contexts, and they are characterized by their Helios^+^ phenotype.

As demonstrated, altered Helios expression in the T lymphocyte pool is linked to an autoimmune phenotype that replicates the typical symptoms observed in SLE, an archetypal condition of systemic autoimmunity. Thus, while the “mystery” surrounding CD8^+^ regulatory T cells is slowly lifting [34], many questions remain, such as what the origin of these KIR^+^/Ly49^+^ CD8^+^ T lymphocytes is, and what is their relationship to other cells in the immune system, like DCs.

### 2.3. Helios in Other Immune Cells: Double Negative (DN) T and NK Cells

#### 2.3.1. Helios in DN T Cells

In addition to CD4^+^ and CD8^+^ T cells, there is a specific subpopulation of T lymphocytes with a TCRαβ^+^ profile that expresses neither CD4 nor the CD8 co-receptor. In terms of their possible origin, the model that currently seems to attract the most interest, according to recent studies, suggests that double negative (DN) T cells are derived from autoreactive CD8^+^ T cells that have lost their CD8 co-receptor [78,79]. In this context, DN T lymphocytes would be characterized by the expression of PD-1 at their surface and by the transient presence of Helios in their nucleus [78]. The pathological role of these PD-1^+^ DN T cells is witnessed by their proinflammatory effector phenotype (CD127^low^) and by the production of the IL-17 cytokine, which is tightly correlated with the autoimmune phenomenon [80,81]. However, the regulatory properties of the DN T repertoire cannot be ignored, and they have also been reported on the other side of the immunological equilibrium, in non-autoimmune contexts and in organ transplantation [82,83,84]. Again, this dual behavior of DN T cells would reflect the complexity of the immune system, the existence of specific subpopulations characterized by the expression of markers yet to be defined, and the influence of the molecular context (autoimmune or non-autoimmune).

The transcriptional profile of DN T cells from mice was recently analyzed [85], revealing different DN T subpopulations to be distinguished by the Helios gene (*Ikzf2*). Indeed, activated and non-activated DN T lymphocytes could be identified through this biomarker. Helios expression is enhanced in naïve DN T cells, whereas activated DN T cells express this transcription factor weakly.

#### 2.3.2. Helios in NK Cells

In humans, after CD4^+^ Tregs and mucosal-associated invariant T (MAIT) lymphocytes, NK cells are the immune cell population with the strongest Helios expression (supplemental Figure S3 from [64]). Nevertheless, there are few data regarding Helios expression in the NK compartment. Helios has been proposed to regulate the activity of hyperreactive NK cells in mice resistant to viral infection [86]. Moreover, Helios downregulation was observed in some subsets of “memory-like” NK cells from cytomegalovirus-infected individuals [87], and finally, the proportion of CD16^+^ CD56^dim^ NK cells was reduced in a lupus patient due to germline mutation of the *IKZF2* gene [64]. Together, Helios appears to be an important transcription factor that controls the activity and homeostasis of NK cells, which are involved in autoimmunity [88] and lupus [89]. Further studies will be needed to examine the expression of Helios in autoimmune contexts and in different immune cells.

## 3. Systemic Lupus Erythematosus (SLE)

### 3.1. SLE Patients

SLE is a multifaceted autoimmune disease with both environmental and genetic influences. It may present in various ways, affecting different organs like the joints, nervous system, skin, kidneys, and heart (Figure 3) [90,91]. One prevalent feature in SLE patients is the presence of circulating antibodies against nuclear antigens, such as dsDNA, nucleosome proteins, and RNA-associated proteins. These antibodies can accumulate in various tissues, provoking damage and promoting an autoimmune response [92]. In general, like other autoimmune diseases, SLE exhibits a cyclic pattern of activity, stability, and occasionally, remission [90,93]. Thus, the diverse manifestations, varying in both form and degree of involvement, lead to heterogeneity in the proposed prevalence rates depending on the method or criteria used to assess patients. Fortunately, more intensive research into this disease is now translating into new, more sensitive, and specific diagnostic criteria for SLE that are being used in clinical practice (EULAR/ACR 2019 criteria vs. ACR 1997 criteria) [94]. SLE shows a clear gender bias, affecting nine females for every one male, making it one of the most extreme gender-biased diseases in medicine [95]. Therefore, finding new reliable markers to diagnose and predict this pathology is an urgent unmet medical need. This achievement would help unify the patient community worldwide and would ideally identify preclinical or incomplete cases [96]. Yet it remains unclear how basic research can address this issue.

### 3.2. Mouse Models of SLE

Many mouse models are available to be used in basic and preclinical research that recapitulate some of the main features of SLE [97,98]: spontaneous, inducible, transgenic, and humanized models. Compared to other spontaneous models, the MRL strain is characterized by (1) the accumulation of a broad repertoire of circulating antibodies against nuclear antigens (anti-dsDNA, anti-Ro, anti-La…), (2) a pronounced sex bias, and (3) by the presentation of additional SLE symptoms like glomerulonephritis, arthritis, nervous system inflammation, skin rashes, and vasculitis. There are two variants of the MRL strain: MRL/LPR mice that present a much more severe pathology with an early onset (at 3–4 months of age), provoked by homozygous mutation in the Fas protein encoding gene (CD95); and MRL/MPJ mice with no such genetic alteration and with a milder pathological phenotype that appears later in life (at 6–12 months of age), reflecting a genetic background prone to the autoimmune phenomenon [99]. Below we will discuss the main alterations in the aforementioned adaptive immune cells in relation to SLE and focusing on the role of Helios.

## 4. Adaptive Immune Disequilibrium in SLE

As we have seen, the immune balance between T cells (regulatory and conventional) and DCs is a complex homeostatic equilibrium that, when disturbed, could explain part of the autoimmune phenomena of the SLE pathology through either a shift towards a loss of tolerance to self-antigens or towards an exacerbated response [100]. Here we shall consider some findings related to the role of these cell types in SLE, with specific reference to the Helios expression in these populations (Figure 3).

### 4.1. CD4^+^ T Cells in SLE

#### 4.1.1. CD4^+^ Tconvs and Tregs in SLE

SLE patients have an expanded population of effector memory CD4^+^ T cells relative to healthy individuals [101]. Thus, the production of IL-2 and IL-17 is altered in this cellular compartment, the two best studied cytokines in the immune balance. Specifically, there is a notorious reduction in IL-2 (essential for maintaining regulatory function) and enhanced IL-17 production (a critical cytokine in promoting autoimmunity), as seen in CD4^+^ Tconv cells from patients and in animal models [102,103]. In addition, there is an expansion of follicular and extrafollicular CD4^+^ T cell populations in SLE, promoting maturation of B cells and antibody release, and correlated with pathological severity [104,105].

There are conflicting data regarding CD4^+^ Tregs in the literature. While some studies demonstrated a smaller proportion of circulating regulatory cells in SLE patients [106], elsewhere, the opposite effect was reported: a significant increase in this immune subpopulation in the periphery as a homeostatic compensatory mechanism aimed at re-establishing the immunological equilibrium [107,108]. This controversy regarding the number and proportion of CD4^+^ Tregs in SLE patients was highlighted in a recent meta-analysis of 18 independent studies [109]. The global analysis revealed a slight decrease in the proportion of CD4^+^ Tregs, accompanied by very strong heterogeneity. It was proposed that the origin of these discrepancies might reside in (1) the criteria used to evaluate and diagnose active and inactive forms of the pathology; (2) the flow cytometry strategy used to define CD4^+^ Tregs; and (3) the treatment type administered to the patients. According to the same meta-analysis [109], there is also no consensus regarding possible defects in the regulatory activity of these cells. While two studies show that CD4^+^ Tregs from SLE patients have an altered immunosuppressive activity [110,111], another study showed no significant difference [112].

#### 4.1.2. CD4^+^ Helios^+^ T Cells in SLE

Against this background, to further investigate the function and phenotype of CD4^+^ Tregs lymphocytes, and to avoid such heterogeneity and disparity of results, an analysis of the Helios transcription factor by flow cytometry has begun to be included in clinical research into autoimmune diseases [77,113]. It is worth noting the consistent increase in the proportion of CD4^+^ Helios^+^ Treg lymphocytes in SLE patients in three independent studies correlated with disease activity [108,114,115]. Because these CD4^+^ Helios^+^ Tregs present a more active immunosuppressive function than CD4^+^ Helios^−^ Tregs, they could represent a component of the immune compensation system and respond to acute immune imbalances that need to be addressed.

### 4.2. CD8^+^ T Cells in SLE

#### 4.2.1. CD8^+^ Tconvs and Tregs in SLE

Although less well-studied than their CD4^+^ counterparts, CD8^+^ T cells are receiving increasing attention as agents involved in the development of autoimmunity, and particularly SLE, due to both their inflammatory and protective functions [103,116,117]. One of the best studied characteristics of CD8^+^ T cells is their ability to induce targeted cell death via the secretion of granzyme and perforin, thereby halting infectious and tumor processes. CD8^+^ T cells from SLE patients exhibit compromised cytotoxicity, as witnessed by the reduced granzyme B and perforin [118]. This phenomenon may explain why viral infections are a leading cause of mortality in SLE patients [119]. In parallel, cytotoxicity is crucial for the optimal regulation of the humoral adaptive response by KIR^+^/Ly49^+^ CD8^+^ Tregs, eliminating autoreactive follicular T lymphocytes. The murine model of SLE in which perforin is lacking provides evidence for this, in which autoimmune symptoms are accelerated [120].

#### 4.2.2. CD8^+^ Helios^+^ T Cells in SLE

As mentioned, the effects of Helios in CD8^+^ T cells have been almost exclusively studied in KIR^+^/Ly49^+^ CD8^+^ Tregs from autoimmune models that mimic SLE. Nevertheless, weaker Helios expression in the CD8^+^ T compartment and smaller numbers of these cells per unit volume have been described recently in the blood of lupus patients [41,121].

### 4.3. DN T in SLE

The association of DN T cells with SLE and with other autoimmune diseases has been well defined [100,118,122,123]. For example, these cells have been implicated in the production of autoreactive antibodies against dsDNA by B lymphocytes in humans [124]. In addition, an increase in the number of DN T cells has also been found in both the peripheral blood and kidneys of patients, and in mouse models of lupus, suggesting their role in tissue damage [125,126]. Also, the proportions of DN T cells were enhanced during autoimmune activity as opposed to remission in young lupus patients [127].

### 4.4. DCs in SLE

As previously discussed, pDCs and cDCs modulate immunogenic and immunosuppressive responses, fulfilling important roles in both innate and adaptive immunity. Therefore, it is not surprising that these cells have received attention for their potential contributions to our understanding of the pathophysiology of SLE [128,129,130]. Overall, the DC repertoire exhibits changes in number, frequency, phenotype, and function as lupus progresses. However, it is difficult to obtain uniform results regarding SLE in clinical practice due to its marked heterogeneity in terms of manifestations and activity. In addition, the different immunosuppressive drugs widely used to treat SLE affect the biology of DCs [131]. Furthermore, it is unclear whether SLE is a result of an intrinsic defect in DCs or if this pathology alters the behavior and phenotype of DCs as a secondary consequence of the molecular context produced mainly by circulating cytokines [132].

#### 4.4.1. cDCs in SLE

Studies analyzing cDC frequencies have produced inconsistent data [128], although most seem to indicate that SLE patients have fewer cDCs in their peripheral blood, which correlates with a stronger infiltration of immature cDCs in the kidneys [133]. The few studies that have focused on the maturation and function of cDCs in SLE have also produced conflicting results. On the one hand, early data seemed to indicate that DCs from SLE patients have fewer co-stimulatory molecules and a weaker capacity to induce T lymphocyte proliferation relative to healthy individuals [134]. By contrast, during active stages of the disease, lower PD-L1 expression and higher CD80 and CD86 expression were detected in cDCs [135].

Due to their scarcity in circulation, information has been gained from DCs differentiated in vitro from monocytes, albeit again with conflicting results. While it was postulated that DCs derived from SLE patient monocytes have a less proinflammatory profile, a reduced capacity to stimulate T cells, and altered maturation [136], elsewhere, DCs were proposed to express CD80 and CD86 more strongly, and with enhanced proinflammatory cytokine secretion [137]. Nevertheless, there seems to be a consensus regarding the decrease in CD83 in the membrane of DCs, a marker of dendritic maturation and activation [138].

#### 4.4.2. pDCs in SLE

One of the most widely accepted models for the initiation of SLE proposes that a malfunction in phagocytosis and in the elimination of circulating apoptotic bodies could lead to their accumulation in organs and tissues [139]. In this context, nucleic acid fragments in this cell debris could activate TLR7 and TLR9 receptors of pDCs, thereby boosting the production and release of type I IFNs into the bloodstream. This cytokine family would activate both the innate and adaptive immune systems, inducing what is known as the IFN signature [140]. The significance of these molecular mediators is evident since over 50% of adult SLE patients—and 90% of affected children—show a marked expression of type I IFN response genes [141]. Indeed, the selective removal of pDCs in SLE patients has beneficial effects on skin symptomatology [142]. Thus, the intricate cellular and molecular interplay among immune cells makes autoimmune diseases a heterogeneous group of pathologies that are difficult to manage. That is why their optimal treatment remains a challenge in contemporary medicine.

## 5. Helios as a Potential Biomarker for SLE: The Link between tolDCs and CD8^+^ Tregs

### 5.1. Helios as a Potential Biomarker in SLE

At the cellular and molecular level, many studies have focused on understanding the homeostatic equilibrium between T cells and DCs in SLE and other autoimmune diseases. Nevertheless, this only represents a small piece in the complete immune system puzzle and many questions remain due to the difficulties in fitting all the pieces together. Resolving these questions could open new opportunities for the treatment of these heterogeneous diseases, among which abundant subclinical cases exist, and new findings are driving constant changes in diagnosis criteria (e.g., in SLE) [143]. Hence, “the search for a specific SLE biomarker” [144] is still one of the most important challenges in relation to this disease. In terms of the ideal SLE biomarker, certain characteristics must be achieved. (1) It must have a clear implication in the pathophysiology of the disease; (2) it must have a high predictive value, specificity, and sensitivity; (3) it must be a biomarker with measurable variation that precedes periods of pathological activity, or in other words, it must be able to monitor the subclinical and incomplete evolution of SLE; (4) finally, this “ideal” biomarker must be reliably identified in biological samples using standardized assays that can be easily reproduced, interpreted, and performed by most laboratories at a reasonable cost [144]. Does Helios fulfill these requirements?

In terms of the first point, Helios could indeed play an essential role in the pathophysiology of lupus, both in humans and in mouse models. First, in mice, the global absence of Helios triggers an autoimmune phenomenon reminiscent of SLE [38,63]. Moreover, in classic murine SLE models like the MRL/LPR and MRL/MPJ mice, there are low levels of Helios in the Ly49^+^ CD8^+^ T cell populations [73]. Very recently, a lupus-like phenotype was diagnosed in a patient with a germline mutation in the *IKZF2* gene [64]. Moreover, a genome-wide association study (GWAS) [145] of more than 7000 European SLE patients identified the *IKZF2* gene as a susceptibility locus. Interestingly, this study identified other transcription factors, including other members of the Ikaros family (*IKZF1* and *IKZF3*), among the candidate predisposition genes for lupus. This may suggest that the risk of SLE is the consequence of alterations to a gene expression network in multiple compartments of the immune system. Regarding the second point, a study in humans demonstrated the predictive value of the IKZF2 gene as a biomarker for the diagnosis of a prevalent SLE manifestation known as lupus nephritis [146]. As all analyses of Helios expression have been undertaken in patients with an evident pathology, its subclinical value is difficult to ascertain. Nevertheless, in asymptomatic MRL/MPJ and MRL/LPR animals (see Section 5.2), differential expression of Helios can be seen prior to disease manifestation and reflecting the future disease course: slow and mild (MRL/MPJ), or fast and severe (MRL/LPR) [73]. Finally, flow cytometry is a well-established experimental procedure in hospitals and immunology laboratories, and it is permanently undergoing advances [147]. Helios could be easily assessed simultaneously in different cell repertoires from a blood sample using this technique.

### 5.2. Helios Expression in T Cell Repertoires from Two Murine Models of SLE

Apart from the aforementioned evidence, a fifth property of Helios could be considered, as its expression follows different dynamics in each T subpopulation. This behavior could provide added value to this transversal T linage biomarker. For example, Helios was expressed more strongly in CD4^+^ Treg and Tconv populations in two murine models of lupus, yet more weakly by γδ T, DN T, and Ly49^+^ CD8^+^ Treg cells during the progression of the disease [73]. Perhaps the Helios^hi^ profile of the CD4^+^ Treg fraction reflects a stronger activation and effector phenotype of these cells, a compensatory mechanism that is unable to combat the autoimmune imbalance. Conversely, this immune disequilibrium could be facilitated by a dampening of the immunosuppressive function of DN T and Ly49^+^ CD8^+^ Treg cells that express Helios weakly.

Remarkably, this biomarker proves to be a better criterion to classify both mouse models (MRL/MPJ vs. MRL/LPR) and disease states (healthy or predisease vs. diseased) relative to traditional criteria widely used in preclinical research, such as the relative proportion of effector memory CD4^+^ T lymphocytes (CD44^+^ CD62L^−^ CD25^−^ CD69^+/−^). For example, the statistical modeling of our data shows that Helios expression in the CD4^+^ Treg fraction is as powerful as anti-dsDNA antibody quantification in blood to distinguish the different groups of mice (Figure 4), an analysis based on a previous study [148]. Regarding the statistical modelling, univariate logistic regression was used to evaluate the predictive ability of the different biomarkers. In cases where complete and quasi-complete separation led to some estimated regression coefficients being infinite, Firth’s univariate logistic regression model was implemented. Forest plots were used to show the odds ratio (OR) with a 95% confidence interval, together with their *p*-values.

### 5.3. Influence of tolDCs in Helios Expression on CD8^+^ T Regs

Trying to connect the aforementioned adaptive elements of this autoimmune puzzle, we found that the molecular and cellular environment could affect Helios expressions by CD8^+^ T populations. In vitro studies on bone marrow-derived DCs revealed that tolDCs were specifically able to boost the expression of Helios in OT-I CD8^+^ T cells in the presence of ovalbumin relative to imDCs and mDCs [73]. This result is in line with previous evidence supporting the link between tolDCs and CD8^+^ Tregs.

In an oncological scenario, an increase in the proportion of CD8^+^ FoxP3^+^ Treg cells has been associated with the presence of tolDCs generated by glucocorticoid exposure [149]. Furthermore, cDC1s were seen to be vital for maintaining peripheral tolerance against self-antigens through their activation of murine CD8^+^ FoxP3^+^ Treg cells [53]. Similarly, CD14^+^ monocytes promote the in vitro generation of immunosuppressive CD8^+^ FoxP3^+^ CD25^+^ Tregs from CD8^+^ CD25^−^ cells in humans [150]. In terms of the potential mechanisms by which tolDCs could affect CD8^+^ T cell activity, the TGF-β produced by tolDCs may be important [151], as it is known to influence Helios expression in CD8^+^ Ly49^+^ Treg cells [41]. Moreover, the lack of DCs in mice leads to a significant decrease in the splenic CD8^+^ CD44^+^ T fraction, which is correlated with an autoimmune phenotype akin to lupus (e.g., splenomegaly, antibodies against circulating nuclear antigens; see [152]). Thus, it could be postulated that the CD8^+^ CD44^+^ T cell population largely corresponds to the CD8^+^ Ly49^+^ Treg cell fraction. If this were indeed the case, DCs would appear to play an integral role in regulating the differentiation and phenotype of a subset of CD8^+^ Ly49^+^ Treg lymphocytes, which helps maintain the immune balance and prevents the onset of systemic autoimmune phenomena. Together, our current understanding of the autoimmune imbalance in lupus supports the use of tolDCs and Tregs as therapeutic agents in cell-based immunotherapies. Nevertheless, this kind of approach still faces several limitations that may well be overcome in the next few years due to promising new advances through approaches like nanotechnology.

## 6. Cell-Based Immunotherapies for Autoimmune Diseases

Systemic and non-specific immunosuppressive treatments used to manage certain autoimmune disorders may have significant side effects, such as cancer and infections, as excessive immunosuppression may arise [153]. In recent years, efforts have been made to develop new immunotherapies targeting specific aspects of the autoimmune pathophysiology without disrupting the entire system. These therapies take advantage of the properties of the immune system itself as a biotechnological tool. In terms of immunotherapies, we will focus on cell-based approaches.

### 6.1. Cell-Based Immunotherapies

As the name suggests, cell immunotherapies are based on the autologous inoculation of immune cells—modified or not—to restore the immune imbalance depending on the disorder. Currently, ex vivo-generated CD4^+^ iTregs [154,155,156] and tolDCs [28,157] are being transferred into patients affected by diverse autoimmune diseases (multiple sclerosis, type I diabetes…) in clinical trials to test their therapeutic efficacy (Figure 5).

In the case of SLE, as there is currently no drug specifically designed to increase Helios levels in target cells, one potential treatment for this pathology could be the injection of therapeutical cells that either express enhanced levels of Helios (CD8^+^ Ly49^+^ iTreg cells) or promote its expression (tolDCs).

### 6.2. Limitations of Cell-Based Immunotherapies

As mentioned previously, DCs regulate adaptive immune responses in lymphoid tissues where they interact with different T cell populations. For successful immunomodulation, inoculated DCs need to overcome several barriers. (1) They must penetrate afferent lymphatic vessels; (2) they must then migrate towards a draining lymph node (LN); (3) and finally, they must be internalized into the LN structure [158]. These limitations would largely be responsible for the poor migratory efficiency of DCs in the target LN (less than 5%) [159]. Given that the magnitude of the T cell response is proportional to the number of injected DCs that reach the LNs [160], strategies to maximize the migration and mobilization of DCs to the LNs are being explored in both autoimmunity and cancer research. This concentration of therapeutic cells at specific targets could avoid the side effects derived from systemic and non-specific dissemination of DCs in the bloodstream. In this context, different approaches are being tested: (1) direct inoculation of DCs into the LNs [161]; (2) induction of enhanced CCR7 expression, a fundamental chemotaxis receptor on DCs [162]; and (3) the magnetic targeting of DCs coupled to iron oxide nanoparticles (IONPs) [163]. This latter strategy will be addressed below as a promising nanotechnology approach to biological problems. However, the therapeutic use of tolDCs for SLE [164] is currently at a preclinical stage, having been successfully tested in vivo in mouse models [165] and explored preliminarily in vitro with human cells [166,167]. However, there are still several obstacles that need to be tackled.

## 7. Nanobiomedicine

Step-by-step medicine is becoming more and more personalized, and nanotechnology is one of the areas revolutionizing biomedicine [168], a Key Enabling Technology recognized by the European Union [169]. The implementation of nanotechnology in biomedicine can be referred to as nanobiomedicine, based on the use of nanometric materials (with at least one dimension between 1 and 100 nm) [170] to diagnose, prevent, monitor, and/or treat disease [171]. At this nanometric scale, near a quantum level, common macroscopic materials could exhibit emerging physical and chemical properties that could be useful in medicine. The current repertoire of available nanoparticles (NPs) is tremendously diverse (e.g., polymeric, metallic, and ceramic NPs), each with unique properties and possibilities [172]. Some examples of metallic NPs include (1) the use of photothermic gold NPs to provoke tumor cell death by heating [173]; (2) the use of silver NPs as antibiotics [174]; (3) the use of magnetic IONPs as a contrast agent for magnetic resonance (MRI), as an inducer of tumor hyperthermia, and as an agent for cellular and molecular targeting (see Section 7.2).

### 7.1. Magnetic Iron Oxide Nanoparticles (MNPs)

As mentioned, magnetic iron oxide nanoparticles (MNPs) are among the best-known and widely studied nanomaterials. Because MNPs can be naturally processed and excreted via the main pathways of iron metabolism, they have been identified as one of the most suitable options for biomedicine in terms of biocompatibility [172,175,176]. Among the most common MNPs considered are MNPs based on magnetite (Fe_3_O_4_), maghemite (γ-Fe_2_O_3_), or mixed ferrites (Co, Mn, Ni, or Zn + Fe_2_O_4_). Overall, IONPs have two main elements: the iron oxide nucleus or core and the chemical coating. On the one hand, the “small” size of the nucleus (<50 nm) favors their superparamagnetic behavior, avoiding exacerbated aggregation of MNPs in the blood when subjected to an external magnetic field and once it is removed. On the other hand, the usually charged chemical coating plays a role in NP design: (1) improving core biocompatibility; (2) reducing immunogenicity; (3) preventing aggregation due to electrostatic repulsion; and (4) acting as an anchor to conjugate other molecules (drugs, antibodies, fluorescent dyes…) [177]. In addition, the coating composition affects the relationship between cells and MNPs. For instance, different coatings drive different types of intracellular trafficking and degradation depending on the cell type [178,179]. Likewise, because the coating has the potential to regulate the quantity of MNPs bound to the cell surface, the choice of coating could be critical to maximize the amount of iron per cell, and thus, their magnetic movement [180].

### 7.2. MNP Applications in Biomedicine

Among the numerous possibilities offered by MNPs in biomedicine, those that stand out include (Figure 6) (1) the localized delivery of molecules with biological activity (drugs, cytokines, antibodies, etc.) by means of magnetic fields to minimize the secondary effects related to their systemic dissemination [181,182]; (2) magnetic hyperthermia, which offers an option for oncological treatment through the heating of MNPs exposed to an alternating magnetic field [183,184,185]; (3) the use of MNPs as contrast agents for MRI [186,187]; (4) the magnetic targeting of immune cells using external magnetic fields as a solution for the previously indicated obstacles to cell-based immunotherapies with Tregs and tolDCs.

#### Magnetic Targeting for Cell-Based Therapies

As indicated above, optimizing therapeutic cell migration to target tissues is key to realizing the full potential of cell-based therapies [188,189] (see Table 3 for examples of magnetic targeting of therapeutic MNPs to cells). Most research into the magnetic targeting of immune cells has focused on oncological therapies. Less attention has been paid to the other side of the immune balance, the potential use of MNPs for the magnetic targeting of immunosuppressive cells (tolDCs, iTregs…) towards lymphoid tissues as a strategy to improve cell-based immunotherapies in autoimmune contexts. What has been studied extensively is the interaction between DCs and different types of NPs [190,191,192], as well as the specific use of nanomaterials to induce the polarization of DCs towards tolDCs in vivo as a therapeutic approach for autoimmune pathologies [193,194]. What lessons can be learned from these studies to extend the idea of magnetic targeting to tolDCs?

### 7.3. MNPs and DCs

Among the vast repertoire of nanomaterials available, the possibility of modifying DCs with iron oxide MNPs has also been analyzed. From these studies, some general lessons were learned, regardless of the biomedical application for which they were intended: (1) at low concentrations (0–50 µg/mL Fe), MNPs do not exert significant cytotoxicity and they do not alter cell morphology, surface marker patterns, or the cytokine profile of previously matured DCs [202,203,204,205,206,207,208,209]. However, some reports did claim that MNPs can potentially reduce the viability of DCs at high concentrations (>400 µg/mL Fe), while not affecting their maturation [210]. In addition, DCs treated with MNPs can maintain a stable phenotype, but they may lose their optimal ability to stimulate CD4^+^ T lymphocytes [211]. In turn, MNPs can experience reduced migratory capacity at high concentrations and when they accumulate at micrometric instead of nanometric levels [206,212]. Regarding their surface electrical charge, positively charged MNPs favor antigen cross-presentation on DCs relative to those with an electrically negative coating [209,213]. This behavior is attributed to the chemical modification of their surface, which dictates their intracellular fate toward a particular compartment [178,179,213]. Also, negatively charged MNPs stimulate stronger production of the proinflammatory IL-1β than positively charged MNPs [209]. Finally, MNP treatment prior to in vitro DC maturation could alter this process [208]. Together, the literature cited suggests the use of positively charged superparamagnetic MNPs at low concentrations (<50 µg/mL) for the magnetic targeting of fully mature tolDCs towards LNs as a new cell-based immunotherapy approach.

## 8. Conclusions

Given the very heterogeneous manifestations of SLE in different cohorts of patients and the presence of abundant subclinical cases, the standard criteria for lupus diagnosis in clinical practice are constantly changing as new findings appear. Hence, the search for better biomarkers for SLE is one of the main challenges in the context of precision medicine for this disease. In this regard, Helios (encoded by *IKZF2*) is a transversal transcription factor that shapes the regulatory activity in the T cell compartment, and currently, it is being investigated intensively in autoimmune contexts. Indeed, its levels are independent and vary in different immune cell populations: DN T cells, CD4^+^ and KIR^+^/Ly49^+^ CD8^+^ T cells, and NKs. The predictive value of Helios in lupus and its role in maintaining the regulatory profile of T cell subpopulations make it a potential candidate to be considered in future studies into SLE classification.

On the other hand, in this review, we have discussed how Helios in CD8^+^ T cells could be influenced by their interaction with DCs, specifically tolDCs. Given that, in preclinical trials of cell-based immunotherapy, tolDCs do not fully restore the immune equilibrium to avoid exacerbated autoimmunity, efforts need to focus on strategies to overcome some of the main limitations associated with this approach. One of these obstacles is the very low proportion of therapeutic cells that reach the target tissue. As a potential solution, the possibility to drive the migration and retention of therapeutic cells by using MNPs and external magnetic fields (magnetic targeting) is currently being studied, with promising results.

## Figures and Tables

**Figure 1 ijms-25-00452-f001:**
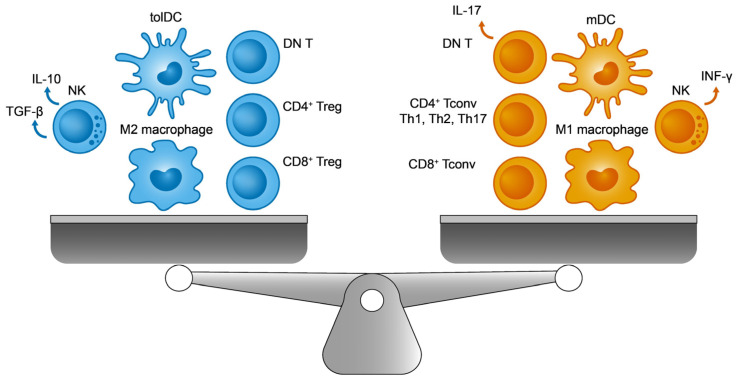
The immune balance. On the (**left**), the immunosuppressive cells include M2 macrophages, regulatory T cells (Tregs), tolerogenic dendritic cells (tolDCs), double negative T cells (DN T), and TGF-β producing natural killer (NK) cells. On the (**right**), their immunogenic counterparts are M1 macrophages, conventional T lymphocytes (Tconv), immunogenic mature DCs (mDCs), IL-17 producing DN T cells, and INF-γ producing NK cells.

**Figure 2 ijms-25-00452-f002:**
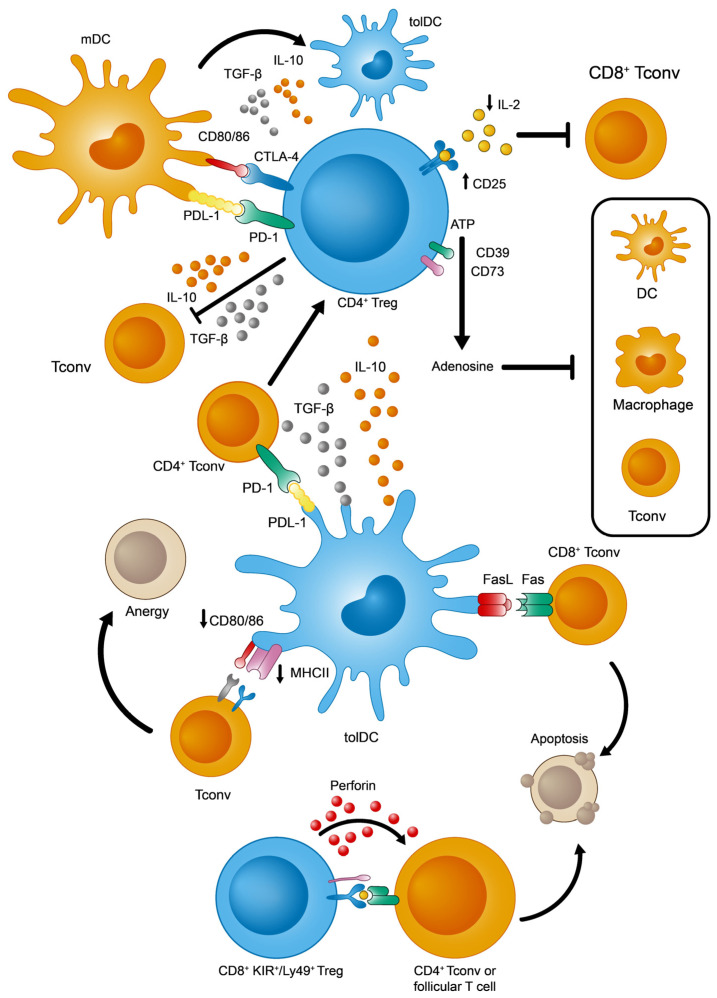
Main immunosuppressive mechanisms adopted by tolDCs, CD4^+^, and KIR^+^/Ly49^+^ CD8^+^ Tregs. Adapted from [24,28,29].

**Figure 3 ijms-25-00452-f003:**
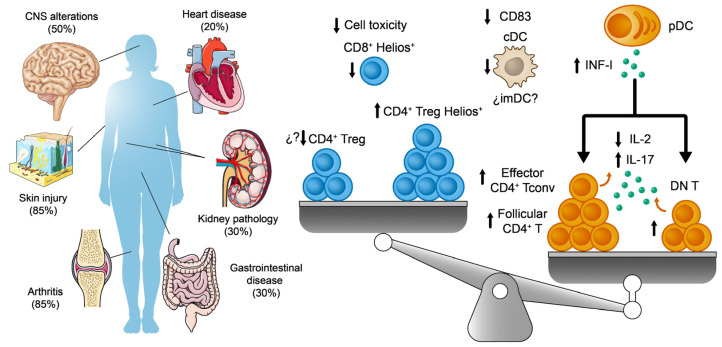
On the left, the main systems affected in SLE are shown. The percentages indicate their prevalence among lupus patients. On the right, a scheme summarizes the major imbalances and alterations to Treg, Tconv, and DC populations associated with SLE.

**Figure 4 ijms-25-00452-f004:**
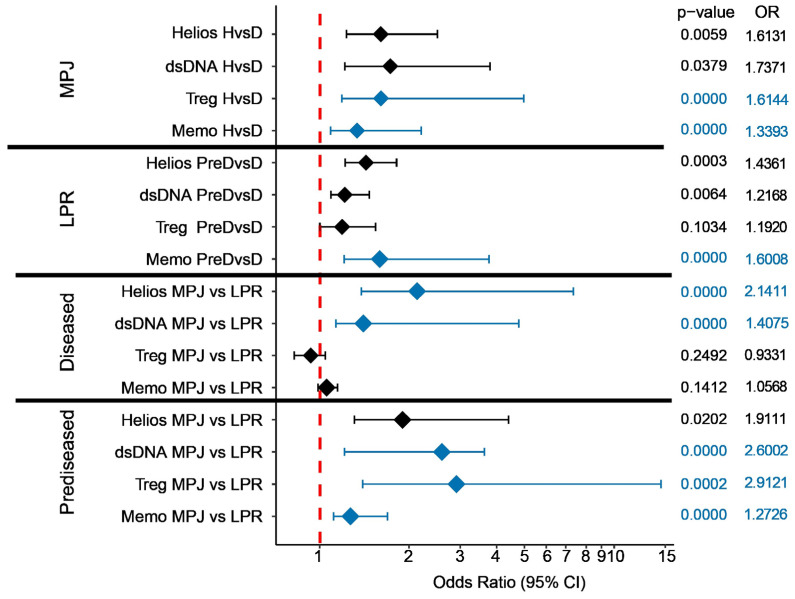
A predictive biomarker analysis in MPJ and LPR mice of two lupus disease states (healthy (H), and prediseased (PreD) or diseased (D)), studying two variants of the mouse MRL background: MRL/MPJ (MPJ) and MRL/LPR (LPR). The data are presented as a Forest plot, with the odds ratios (ORs) and 95% confidence interval (95% CI) assessed by a univariate (black) or Firth’s univariate (blue) logistic regression. Unit changes for the continuous variables were assessed for the logistic regression. For circulating antibodies against double-stranded DNA molecules (dsDNA), their biological activity was divided by 10,000. Helios represents the proportion of Helios^+^ cells in the CD4^+^ Treg fraction; Treg, the proportion of Treg (FoxP3^+^CD25^+^) cells in the CD4^+^ compartment; and Memo, the proportion of CD44^+^ CD62L^−^ CD25^−^ cells in the CD4^+^ population. The data presented are derived from a previous study [73], and in addition, new flow cytometry analyses were taken into account. The analysis was performed and the graph obtained using the stats, logistf, and ggplot2 packages of R version 4.2.2.

**Figure 5 ijms-25-00452-f005:**
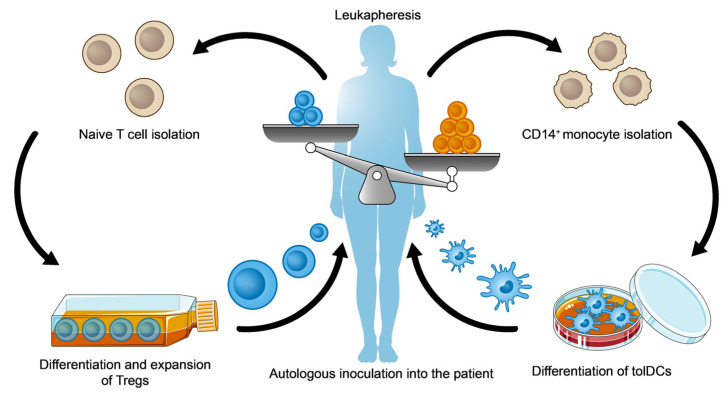
Summary of two of the main strategies currently being tested in clinical trials to restore the altered immune balance in autoimmune diseases using cell transfer-based immunotherapy involving iTregs and tolDCs.

**Figure 6 ijms-25-00452-f006:**
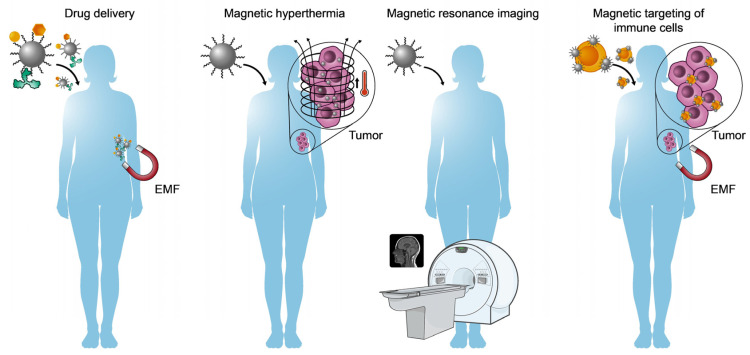
Biomedical applications for magnetic iron oxide nanoparticles (MNPs). From the left, drug delivery via external magnetic fields (EMFs), cancer cell hyperthermia using MNPs and alternating magnetic fields, MNPs as contrast agents for magnetic resonance imaging, and magnetic targeting of immune cells with MNPs to optimize immunotherapy through EMFs.

**Table 1 ijms-25-00452-t001:** Main human and mouse markers for different T cell subsets in terms of the activation of memory status.

	Human	Mouse
Naïve	CD45RA^+^ CD45RO^−^ CD25^−^ CD127^+^	CD44^−^ CD62L^+^ CD25^−^ CD69^−^
Effector	CD45RA^+/−^ CD45RO^+/−^ CD25^+^ CD127^−^	CD44^+^ CD62L^−^ CD25^+^ CD69^+^
Central Memory	CD45RA^−^ CD45RO^+^ CD25^+^ CD127^+^	CD44^+^ CD62L^+^ CD25^−^ CD69^+/−^
Effector Memory	CD45RA^−^ CD45RO^+^ CD25^−^ CD127^+^	CD44^+^ CD62L^−^ CD25^−^ CD69^+/−^

**Table 2 ijms-25-00452-t002:** Main human and mouse markers for plasmacytoid (pDC), conventional type 1 (cDC1), and conventional type 2 (cDC2) dendritic cells.

	Human	Mouse
pDC	BDCA2^+^ BDCA4^+^ B220^+^	PDCA-1^+^ B220^+^
cDC1	XCR1^+^ BDCA3^+^	XCR1^+^ CD8α^+^
cDC2	BDCA1^+^ CD11b^+^	CD11b^+^

**Table 3 ijms-25-00452-t003:** Some successful examples of magnetic targeting of therapeutical cells.

Therapeutical Cell Type	Human	Mouse
Endothelial cell	Rat carotid arteries	[195]
Immunogenic mDCs	Murine lymph nodes	[163]
Stem cells	Rat spinal cord lesions	[196]
Murine primary CD8^+^ T cells	Murine lymph nodes	[180]
Murine primary CD8^+^ T cells	Solid tumor	[197]
Human NK-92MI cell line	Murine tumor tissue	[198]
Human primary T cells	In vitro	[199]
Human T cell line (Jurkat)	In vitro	[200]
Mouse T cell line (EL4)	In vitro	[201]

## Data Availability

Not applicable.

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
