# Peer review of "Helios as a Potential Biomarker in Systemic Lupus Erythematosus and New Therapies Based on Immunosuppressive Cells"

_ijms, 2023, doi:10.3390/ijms25010452_

Round 1

Reviewer 1 Report

Comments and Suggestions for Authors

The manuscript entitled “Helios as a potential biomarker in systemic lupus erythematosus and new therapies based on immunosuppressive cells” is a comprehensive review on the role of Helios on the function of different cell types of the Immune System with a special focus on its role in the context of the autoimmune disease Systemic Lupus Erythematosus (SLE), either in humans or in experimental models of the disease. Based in that knowledge the authors propose the use of Helios as a potential biomarker of the progression of the disease. However, the only published paper on that subject in humans propose the use of IKZF2 as a biomarker associated with lupus nephritis (reference 146 of the manuscript), not with other manifestations of the disease. On the other hand, the authors provide useful information on the potential use of Helios expression on different T cells to better classify 2 experimental models of lupus, along the different states of the disease (healthy or pre-disease vs. diseased).

The second part of the review has to do with a completely different subject, “New therapies based on immunosuppressive cells”, which in my opinion could be part of a second manuscript, rather than the continuation of the review on Helios. There is not much connection between the 2 parts of the review, and in fact Helios is not even mentioned in the second part of the review. The inclusion of the second part makes the review lengthy and somehow out of focus.

Author Response

We would like to thank Reviewer 1 for these useful comments. As it is mentioned, the reference 146 cited in the manuscript strictly establish a link between IKZF2 gene expression and lupus nephritis. This meta-analysis was performed in glomerulus and tubulointerstitial tissue samples from lupus nephritis patients by comparing them with healthy donors. As it is stated in the main text: 

 “However, due to the ethical and traumatic reasons, SLE patients without nephritis may not be punctured by the kidney. We cannot compare patients with nephritis against patients without nephritis.” 

Thus, the data from SLE general patients without lupus nephritis is not available. Accordingly, we propose to rewrite the line 517-519 (see in the revised version of the manuscript). 

Old line: “Regarding the second point, a study in humans demonstrated the predictive value of the IKZF2 gene as a biomarker for the diagnosis of lupus nephritis [146].” 

New line 517-519: “Regarding the second point, a study in humans demonstrated the predictive value of the IKZF2 gene as a biomarker for the diagnosis of a prevalent SLE manifestation known as lupus nephritis”.  

Indeed, authors suggest including IKZF2 as new index for lupus nephritis evaluation. 

“It’s possible that IKZF2 might become a new index for prediction and evaluation of LN.” 

Thus, in this part of the review we propose to extend Helios analysis to different immune cell populations in blood to globally study lupus progression in all cases. For example, Helios analysis in Treg populations (in three independent studies: references 108, 114 and 115) demonstrated correlation with disease activity. 

On the other hand, we also propose to introduce cell-based immunotherapies by linking this second part with the fact that there is no available drug designed to boost Helios levels on target cells. We would include a new paragraph on line 605. 

“In the case of SLE, as there is currently no drug specifically designed to increase Helios levels in target cells, one potential treatment for this pathology could be the injection of therapeutical cells that either express enhanced levels of Helios (CD8+ Ly49+ iTreg cells) or promote its expression (tolDCs).” 

Reviewer 2 Report

Comments and Suggestions for Authors

I considered the manuscript entitled “Helios as a potential biomarker in systemic lupus erythematosus and new therapies based on immunosuppressive cell” by Andrés París-Muñoz, et al, that is intended to be published in International Journal Molecular Sciences.

I really enjoyed the manuscript. It has a clear wording, a fantastic organization, a lot of interesting and well-described data, and it offers a new biomarker such as Helios. I have no objection to its publication.

Author Response

Thank you very much for the constructive comments you have made.

Reviewer 3 Report

Comments and Suggestions for Authors

Helios as a potential biomarker in systemic lupus erythematosus and new therapies based on immunosuppressive cells

Knowledge of the immune system and its mechanisms is essential for a true comprehension of autoimmune diseases. Therefore, to be able to understand the pathogenesis of systemic lupus erythematosus (SLE) and the immune disbalance that leads to its clinical manifestations, one must be aware of the elements that keep this fragile equilibrium. Such elements are represented by conventional CD4+ and CD8+ T cells, regulatory CD4+ T and CD8+ cells, tolerogenic dendritic cells. In this immunological expanse, Helios seems to possess a regulatory quality. Helios is a transcription factor, member of the Ikaros family of transcription factors, that may influence Natural Killer (NK) cells activity and maintain the regulatory profile of T-cells. Thus, Helios is an essential factor in balancing immune processes, and, as such, in autoimmunity control. The review is relevant for future studies, as it brings forth the necessity of thorough understanding of autoimmunity and its consequences in order to truly comprehend diseases such as SLE, but also for the potential of Helios as a biomarker in diagnosing SLE. Particularly in such a debilitating disease, that is, at times, difficult to diagnose due to its heterogenous manifestations, such a marker would be immensely helpful in an early diagnosis, and also to monitor disease activity in a subclinical state.

The manuscript presents a logical structure and a high level of written clarity. The difficulty of the notions presented have a gradual progression throughout the text, an essential element given the complexity of the subject. The manuscript has quality scientific content and is optimally researched and referenced. The use of tables and complex figures further increases the ease of comprehension of the text. One of the highlights of the review is the eloquent analysis of novel treatment prospects, that encompasses a promising perspective for patients with autoimmune diseases.

After analyzing this manuscript, it can be considered for publication.

Author Response

(The authors gave the same response as above.)

Round 2

Reviewer 1 Report

Comments and Suggestions for Authors

No further comments